# Immune Checkpoint Inhibitor Treatment and Ophthalmologist Consultations in Patients with Malignant Melanoma or Lung Cancer—A Nationwide Cohort Study

**DOI:** 10.3390/cancers14010049

**Published:** 2021-12-23

**Authors:** Maria D’Souza, Mette Bagger, Mark Alberti, Morten Malmborg, Morten Schou, Christian Torp-Pedersen, Gunnar Gislason, Inge Marie Svane, Jens Folke Kiilgaard

**Affiliations:** 1Department of Ophthalmology, Copenhagen University Hospital Rigshospitalet, 2100 Copenhagen, Denmark; mette.marie.bagger.01@regionh.dk (M.B.); mark.jensen.alberti@regionh.dk (M.A.); jens.folke.kiilgaard@regionh.dk (J.F.K.); 2Department of Cardiology, Copenhagen University Hospital Herlev-Gentofte, 2900 Gentofte, Denmark; mortenmalmborg@gmail.com (M.M.); morten.schou.04@regionh.dk (M.S.); gunnar.gislason@regionh.dk (G.G.); 3The Danish Heart Foundation, 1127 Copenhagen, Denmark; 4Department of Clinical Medicine, University of Copenhagen, 2200 Copenhagen, Denmark; inge.marie.svane@regionh.dk; 5Department of Public Health, Copenhagen University, 1353 Copenhagen, Denmark; christian.torp-pedersen.01@regionh.dk; 6Department of Cardiology, Copenhagen University Hospital Hillerød, 2400 Copenhagen, Denmark; 7Department of Cardiology, Aalborg University Hospital, 9100 Aalborg, Denmark; 8Department of Oncology, Copenhagen University Hospital Herlev-Gentofte, 2730 Herlev, Denmark

**Keywords:** ocular inflammation, uveitis, malignant melanoma, lung cancer, immune checkpoint inhibitors, one-year risk, epidemiology

## Abstract

**Simple Summary:**

Immune checkpoint inhibitors are increasingly being used for treating advanced malignant cutaneous melanoma and lung cancer. Immune-related side effects in multiple organs are common but the frequencies of ophthalmic side effects in national cohorts of unselected patients are undescribed. This study estimated frequencies of first-time ophthalmologist consultations and inflammatory conditions in consecutive patients with malignant melanoma or lung cancer treated with immune checkpoint inhibitors in Denmark from 2011–2018. The one-year risks of first-time consultation and ocular inflammation were 6% and 1%, respectively. These numbers were increased compared with patients with the same type of cancer who were not treated with immune checkpoint inhibitiors.

**Abstract:**

Purpose: To estimate the frequency of first-time ocular events in patients treated with immune checkpoint inhibitors (ICI). Methods: Patients with cancer in 2011–2018 in Denmark were included and followed. The outcomes were first-time ophthalmologist consultation and ocular inflammation. One-year absolute risks of outcomes and hazard ratios were estimated. Results: 112,289 patients with cancer were included, and 2195 were treated with ICI. One year after the first ICI treatment, 6% of the patients with cancer, 5% and 8% of the lung cancer (LC) and malignant cutaneous melanoma (MM) patients, respectively, had a first-time ophthalmologist consultation. The risk of ocular inflammation was 1% (95% confidence interval (CI) 0.4–1.2). Among patients with MM, ICI was associated with ocular inflammation in women (HR 12.6 (95% CI 5.83–27.31) and men (4.87 (95% CI 1.79–13.29)). Comparing patients with and without ICI treatment, the risk of first-time ophthalmologist consultation was increased in patients with LC (HR 1.74 (95% CI 1.29–2.34) and MM (HR 3.21 (95% CI 2.31–4.44). Conclusions: The one-year risks of first-time ophthalmologist consultation and ocular inflammation were 6% and 1%, respectively, in patients treated with ICI. In patients with LC and MM, the risk was increased in patients with ICI compared with patients without ICI.

## 1. Introduction

Treatment with immune checkpoint inhibitors (ICI) may dramatically improve the survival in patients with malignant cutaneous melanoma, non-small cell lung cancer, Hodgkins’ lymphoma, urothelial carcinoma, squamous cell carcinoma, and renal cell carcinoma [1,2,3,4]. In the last decade, the indications for ICI treatments have increased exponentially. Of all patients with cancer in the USA, 1.54% in 2011 were estimated to be eligible for ICI treatment, and the number increased to 44% in 2018 [5]. Currently, ICIs approved for clinical use are antibodies functioning as inhibitors of cytotoxic T-lymphocyte-associated protein 4 (CTLA-4), programmed cell death protein 1 (PD1), and programmed death-ligand 1 (PD-L1). CTLA-4 is a transmembrane protein expressed on the surface of activated immune cells, e.g., T-cells. Binding to CTLA-4 produces inhibitory signals to CD8+ T cells involved in controlling cancerous cells. PD1 is a surface receptor involved in regulating the exhaustion and tolerance of mainly T-cells. Inhibiting these immune checkpoints with ICI therapy increases the immune system response primarily via CD8+T cells and thus counteracts the tumor cells immune system evasion [6,7,8]. As the treatment targets key regulators of the immune system, it comes with a high risk of immune-related side effects. Mild (Grade 1–2) side effects are seen in 40% and severe (Grade 3–4) in 2% of patients with mixed cancer indications [9,10]. In patients with advanced lung cancer, the numbers are approximately 9% and 1% [11]. Multiple organs may be involved, most commonly the colon, lungs, and skin [10,12,13]. The current American Society of Clinical Oncology Clinical Practice guideline recommendation is to treat aggressively with glucocorticoids and to consider discontinuation of ICI at Grade 2–4 side effects [14]. Case series and adverse event studies of uveitis and other ocular inflammations after ICI treatment have been published [15,16,17,18]. Frequencies of ocular side effects, including inflammatory ocular side effects, requiring ophthalmologist consultations are undescribed in nationwide unselected cohorts and potentially higher than expected. In this study, we aimed to quantify the risk of first-time ophthalmologist consultation and ocular inflammation associated with ICI in a national cohort of Danish patients from 2011–2018. We found that ICI treatment with inhibitors of CTLA-4 (ipilimumab), PD1 (pembrolizumab and nivolumab), or PD-L1(atezolizumab and durvalumab) was associated with increased relative rates of both ocular inflammation and ophthalmologist consultations at secondary and tertiary hospitals.

## 2. Methods

### 2.1. The Danish Health Care System

In Denmark, health care including oncological and ophthalmological treatment is tax-financed and accessible to all citizens via public health insurance. All Danish citizens are identified via a unique permanent personal social security number given at birth or immigration. The number is filed at all health care contacts and prescription reimbursements and registered within the national Danish administrative registers [19,20]. Thus, the national administrative registers are perceived to be complete registers of Danish health care activities and the social security numbers allow for cross-linkage between the registers and enable follow up back and forth in time on an individual level.

### 2.2. Data Sources

The study was based on data from the national Danish administrative registers. Cross-linkage between the registers was performed via the unique permanent personal security number that all Danish citizens have. Information on date of birth and immigration and emigration status was collected from the Danish Civil Registration System [21]. Date of death was collected from the Danish Register of Causes of Death and information on redeemed prescriptions was collected from the Danish Register of Medicinal Products [19,22]. Data on health care contacts including admission and discharge dates and diseases diagnosed and treated were retrieved from The Danish National Patient Register [20].

### 2.3. Study Population

Patients diagnosed in 2011–2018 with a cancer where a potential ICI indication was present were included. Patients with previous uveal or ocular malignant cutaneous melanoma were excluded (*n* = 29, ICD 10 code: C69 (ocular cancer), C431 (malignant melanoma on eyelids)).

A sub cohort consisting of the patients treated with ICI was analyzed for 30 days and 1-year risks of ocular events and first ophthalmological contacts.

### 2.4. Cohorts

For the analyses of relative rates, two different follow up analyses were conducted—one for each outcome. For the analysis of the outcome of the first ophthalmologist consultation, the patients were followed until they (1) had a first ophthalmologist consultation, (2) died, (3) emigrated, or (4) were alive and free of outcome on December 31st, 2018, where the study period ended. For the analysis of the outcome of ocular inflammation, the patients were followed until they (1) had a diagnosis of ocular inflammation or (2) died, (3) emigrated, or (4) were alive and free of outcome on 31 December 2018, where the study period ended.

In all analyses, patients who developed incident uveal or ocular malignant melanoma during follow up were censored. This censoring criterion was used to secure that the initial ophthalmologist consultation was not due to an ocular cancer primarily diagnosed and treated by specialists in ophthalmology.

### 2.5. Treatment with Immune Checkpoint Inhibitors

Initial ICI administration was defined as having a procedure code with administration of ipilimumab, pembrolizumab, nivolumab, atezolumab, or durvalumab (procedure codes BOHJ19D, BOHJ19J3, BOHJ19H2, BOHJ19J2, and BOHJ19H7) during a hospital contact. Only first-time registrations were included. The method has recently been validated for chemotherapy in colon cancer with positive predictive values of 0.91 (95% confidence interval (CI) 0.90–0.92) [23].

### 2.6. Ophthalmologist Consultation and Ocular Inflammation

The outcome initial ophthalmologist consultation was defined as a first-time hospital contact at an ophthalmological department. The outcome ocular inflammation was defined as a first-time hospital contact with a primary diagnosis of uveitis, conjunctivitis, scleritis, keratitis, or retinitis (ICD 10 codes H20, H10, H15, H16, and H30).

### 2.7. Comorbidities

Relevant comorbidities were defined from admissions to hospital or outpatient treatment with the diagnosis codes listed in the Appendix A. Comorbidities with diabetes mellitus, inflammatory arthritis, hypertension, chronic kidney disease, connective tissue disease, sarcoidosis, multiple sclerosis, or borrelia infections were registered within 5 years before the study entry. Any registration with morbus Bechterew, juvenile arthritis, human immune deficiency virus infection, or syphilis before having the cancer diagnosis at study entry was defined as comorbidity.

### 2.8. Statistical Methods

All analyses were on incident first-time outcomes, e.g., in the analysis of ophthalmologist consultation, only patients without previous ophthalmologist consultations were included. This meant that in the absolute risk analyses, patients with outcomes before ICI were excluded, and in the analyses of relative rates, patients with outcomes before cancer diagnosis were excluded.

The risk time (time from initial ICI administration to event) was summarized in medians with 25 and 75 percentiles (p25–p75).

The Aalen–Johansen estimator with competing risk of all cause death was used for estimating absolute risks at 30 days, 6 months, and 1 year after initial ICI administration. Kaplan–Meier estimates were used for the risk of all cause death at 30 days, 6 months, and 1 year.

Hazard ratios were modeled in multivariable Cox regression models. For each of the sub cohorts (all patients with cancer, patients with lung cancer, and patients with malignant cutaneous melanoma, respectively) a cox model was used for analyzing each outcome (ocular inflammation, uveitis, and first-time ophthalmologist consultation, respectively). ICI exposure was included as a time-updated variable. The dataset was split on the date of first ICI administration and a proxy variable was created, enabling an analysis where patients contributed to risk time in the non-exposed group from cancer diagnosis to the date before first ICI administration and to risk time in the exposed group from the date of first ICI-exposure to end of follow up. Likewise, age and calendar time were included as time-updated variables. Age was included as categorical variable with the levels ≥25, 26–50, 50–75, and >75 years of age. In modeling ocular inflammation, linearity could not be assumed for the age variable. For this reason, age was not included as a variable in these models. Calendar time was included as years since study entry, which was equal to years since cancer diagnosis. Additionally, the models were adjusted for sex. In the Cox model analyzing the association between ocular inflammation and ICI in patients with malignant cutaneous melanoma, interaction with sex could not be ruled out, and an interaction term with ICI treatment and sex was included.

The statistical analyses were performed using SAS Software version 9.4 (SAS Institute Inc.) and R: A language and environment for statistical computing (version R-4.0.3) [24].

## 3. Results

In total, 112,260 patients with cancer were included, 29,337 with lung cancer, 16,023 with malignant cutaneous melanoma, 6526 with head and neck cancer, 7133 with urinary tract cancer, 45,661 with skin cancer, 6493 with kidney cancer, and 1087 with Hodgkin’s lymphoma (Figure 1). At cancer diagnosis, the median age was 69 (p25–p75: 60–78) years, and 53% were men. The most frequent comorbidities were hypertension (33%) and diabetes mellitus (9.7%). Baseline comorbidities with chronic tissue disease and inflammatory arthritis were less frequent (1.5% and 3.5%, respectively) (Table 1).

### 3.1. Initial ICI Treatment

Following the cancer diagnosis, 2190 received treatment with ICI. The initial treatment was ipilimumab (10.9%), pembrolizumab (36.7%), nivolumab (43.2%), atezolizumab (3.3%), durvalumab (0.3%), and ipilimumab combined with nivolumab (5.6%) (Table 2). The time from cancer diagnosis to initial ICI administration was median 338 days (p25–p75 110–746) and varied between cancer types (Appendix A). The most frequent cancer types in patients treated with ICI were lung cancer (54.2%) and malignant cutaneous melanoma (25.1%). Median age was 67 years (p25–p75 59–73) and 56.4% were men. The most frequent comorbidities were hypertension (27.9%) and diabetes mellitus (10.4%).

### 3.2. The Risk of Ophthalmologist Consultation and Ocular Inflammation

One year absolute risks of first-time ophthalmologist consultation after initiation of ICI were 6%, 5%, and 8% in patients with cancer, lung cancer, and malignant cutaneous melanoma, respectively (Table 3 and Figure 2). One year risks of ophthalmological consultation stratified on drug type were 16.3 % (95% CI 8.1–24.5) in patients treated with ipilimumab combined with nivolumab, and 8.7% (95% CI 4.8–12.7), 5.7% (95% CI 3.7–7.5), and 5.0% (95% CI 3.0–7.0) in monotherapy with ipilimumab, nivolumab, and pembrolizumab, respectively (Appendix A). One year absolute risks of ocular inflammation after ICI were 0.8% (95% confidence interval (CI) 0.4–1.2) in all patients and 1.9% (95% CI 0.7–3.2) in patients with malignant cutaneous melanoma.

Median times from first ICI to initial ophthalmologist consultation were 108 days (p25–p75: 53–223) in patients with cancer (all types) and 116 days (p25–p75: 69–222) and 139 days (p25–p75: 54–245) in patients with lung cancer and malignant cutaneous melanoma, respectively. The time from the initial ICI treatment to an incident ocular inflammation diagnosis was median 140 days (p25–p75: 69–250) for all patients with mixed cancer types, 187 days (p25–p75: 138–327) in patients with lung cancer, and 135 days (p25–p75: 63–272) in patients with malignant cutaneous melanoma (Table 4).

In total, 108 out of 2190 patients treated with ICI had a first-time ophthalmologist consultation during ICI therapy. The most frequent diagnoses from these consultations were cataract (*n* = 22, 20%), uveal disease (*n* = 10, 9%), chorio-retinal disease (*n* = 10, 9%), visual loss (*n* = 8, 7%), or miscellaneous (*n* = 29, 27%). None of the 108 consultations were planned control visits in patients with comorbid diabetes mellitus or juvenile arthritis.

Among patients with lung cancer, the patients with ICI treatment had an increased relative rate of first-time ophthalmologist consultation (HR 1.74 (95% CI 1.29–2.34) compared with patients without ICI treatment (Figure 3). The ICI receivers with lung cancer did not have higher rates of ocular inflammation compared with patients without ICI treatment (HR, 2.33 (95% CI 0.70–7.79). Regarding patients with malignant cutaneous melanoma, the ICI-treated patients had increased relative rates of first-time ophthalmologist consultation (HR 3.21 (95% CI 2.31–4.44) compared with patients without ICI treatment. In patients with malignant cutaneous melanoma, ICI was associated with ocular inflammation, and the association was stronger in women. The relative rate of ocular inflammation was increased in patients with ICI compared with patients without ICI in both women (HR 12.6 (95% CI 5.83–27.31)) and men (4.87 (95% CI 1.79–13.29).

## 4. Discussion

The current study presents two main findings related to the ocular risk after ICI treatment. First, among unselected consecutive ICI-treated patients nationwide in Denmark, the one-year risk of first-time ophthalmologist consultations at secondary and tertiary hospitals and ocular inflammation reached 6% and 1%. Second, in patients with lung cancer, ICI was associated with higher relative rates of first-time ophthalmologist consultation but was not associated with ocular inflammation. In patients with malignant cutaneous melanoma, ICI was associated with higher rates of both first-time ophthalmologist consultation at a secondary or tertiary hospital and ocular inflammation.

### 4.1. One-Year Risks of First-Time Ophthalmologist Consultations and Incident Ocular Inflammation

The current findings of first-time ophthalmologist consultation reaching a one-year risk of 4–6% in patients treated with ICI monotherapy appears substantially higher than expected from the irAEs observations from the clinical trials and pharmacovigilance studies presented above. The high estimates suggest that incident ocular symptoms demanding specialist assessment in a secondary or tertiary hospital are much more frequent in the clinical setting than previously assumed. Moreover, our numbers support that the risks of ocular symptoms requiring specialist attention are more frequent in patients treated with dual immune checkpoint inhibition with ipilimumab and nivolumab than in patients treated with monotherapy with ipilimumab or nivolumab.

We found an absolute one-year risk of ocular inflammation of 1% in all patients and 2% in patients with malignant cutaneous melanoma and a 0.5% uveitis risk in all patients. In clinical trials of ICI versus placebo, the incidence of uveitis as an immune related adverse event (irAE) ranged from 0.3% to 6% with the highest risk observed in combination therapy with nivolumab and ipilimumab [4,25,26,27]. A pharmacovigilance study of irAEs found a 3% proportion of ocular irAEs out of all irAEs in patients treated with ICI [28]. Additionally, they found an increased risk of reporting uveitis compared with other irAEs. A French pharmacovigilance study found an incidence of 0.7 cases of moderate-to-severe ocular adverse events per 1000 patient-months of treatment [29].

### 4.2. Relative Rates

We found increased rates of first-time ophthalmologist consultations and ocular inflammations in patients treated with ICI compared with patients without ICI treatment. The increased relative rates were found in both the cohorts of (1) patients with lung cancer and (2) patients with malignant cutaneous melanoma, respectively. These results confirm the findings from a meta-analysis of clinical trials showing increased risk of all-grade immune-related ocular toxicities in patients treated with ICI compared with patients receiving placebo (odds ratio 3.40 [95% CI: 1.32–8.71; *p* = 0.01]) [25]. The findings of increased rates of ocular side effects associated with ICI treatment are supported by pharmacovigilance study data. In a large study based on data from U.S. FDA’s Adverse Events Reporting System (FAERS) database from 2003 to 2018, the rates of ocular side effects in patients treated with immune checkpoint inhibitors were increased compared with rates in patients treated with other drugs [30]. In the current analyses, we found a sex difference when analyzing data from the cohort of patients with malignant cutaneous melanoma. In the subgroup of females in this cohort, ICI treatment had a stronger association with ocular inflammation. Thus, the estimated HR was 12.6 (95% CI 5.83–27.21) in women and 4.87 (95% CI 1.79–13.29) in men. A large meta-analysis based on data from 11 randomized controlled studies including 4965 study participants in total supports the finding of a sex difference in the ocular side effect risk associated with ICI treatment. The study finds larger proportions of ocular irAEs reported in patients with female sex and malignant cutaneous melanoma [25].

Reviews of case series suggest that all ocular tissues bear potential for ICI related inflammation. Reports of conjunctivitis, keratitis, uveitis, scleritis, and retinitis exist, and a wide spectrum of inflammatory response from mild to severe is described [16,18,31,32,33,34]. Our findings match this idea, as we observe low numbers of specific diagnoses compared with the very high number of first-time ophthalmologist consultations. The results may mirror a wide spectrum in the severity of ocular symptoms. In addition, dry eye syndrome has been reported as the most frequent adverse event and would not have been included in the specific diagnoses composing the ocular inflammation outcome definition used in this study [25].

The typical onset time of ocular side effects remains an open question. We observed a large range of latency with 25 and 75 percentiles of 53–368 days after the first ICI administration. Median values were 102–206 days. As was visible in the plots of cumulative incidence, the risks of ocular inflammation and ophthalmological contacts were increasing steadily over the course of the first year. These are novel findings as most other case studies describe shorter latency from ICI administration to the debut of eye symptoms in the majority of patients [15,16,25].

The substantial mortality risk in this group of patients with cancer should be taken into account when interpreting the current findings. As visualized in Figure 2 (the cumulative incidence plots), the one-year risk of death in this group of patients with cancer is high and reaches approximately 50% after one year. In comparison, the ocular risks are small but may very well represent an important burden to both the patients and health care providers in the ICI treatment period. Case series suggest that the majority of ocular side effects may be managed with topical therapy and do not require discontinuation of the ICI treatment [35]. It is important for oncologists to be aware of potential eye symptoms and to know the potential for treatment from ophthalmological specialist examination and care.

### 4.3. Strengths and Limitations

The study was based on nationwide data including consecutive unselected patients. The follow up was complete, and events were not restrained to reported irAEs. The completeness and nature of the data were great strengths of this study. Importantly, the study presents data that are independent from trial registrations and adverse event reports. Hence, it adds to the current published knowledge from clinical trials and pharmacovigilance studies.

We did not have information on cancer stage and associated previous or concomitant treatments, clinical measures of visual acuity, or ophthalmological findings. This information would have contributed to the analyses but was impossible to collect in the study design used. Additionally, we did not have data on the treatment regime after the initial administration. We suspect that the majority of patients were treated with standard regimens. The group of patients who had other-than-standard regimens would indeed be interesting to study in future projects.

Some randomized controlled trials testing oncological treatments may include specified ophthalmological control visits. These visits are specified with a procedure ICD-10 code (ZZ0152). Of the patients treated with ICI and included in this study, 46 had a protocolled ophthalmological control. None of the patients with events (ocular inflammation or ophthalmological contact) had a protocolled ophthalmological visit, and two of the patients with ocular inflammation before ICI and nine of the patients with ophthalmological contacts before first ICI administration had protocolled visits. The patients experiencing events before first ICI-administration were, as described in the methods section, not included in the analyses of the relevant same events.

The outcomes ocular inflammation and uveitis were defined from diagnoses at discharge from a Danish hospital and may be subject to some misclassification bias. However, most definitions of outcomes based on discharge codes from the NDPR have high positive predictive values [20]. The outcome ophthalmologist consultation was defined from contacts with an ophthalmological hospital clinic and did not include contacts with private practice ophthalmologists. Most likely, the definition we used underestimated the outcomes of ocular inflammation and ophthalmologist consultations, as the private practice contacts were left out. Patients presenting with severe eye symptoms may be more likely to be handled within the hospital settings, but patients with mild ocular symptoms may be referred from their oncologist to a private practice ophthalmologist. In this study, we were not able to estimate the frequency of these contacts. If patients with cancer treated with ICI are being referred to private practice ophthalmologists more often than patients with the same type of cancer without ICI treatment, a potential surveillance bias may be present. Future studies on this type of ophthalmological contacts would be valuable.

## 5. Conclusions

In conclusion, ophthalmological side effects to ICI may be more frequent than previously estimated and further investigations of the associated prognosis and treatment are needed in these patients. This study quantified high one-year risks and increased relative rates of ocular inflammation and initial ophthalmologist consultations associated with ICI treatment in patients with cancer.

## Figures and Tables

**Figure 1 cancers-14-00049-f001:**
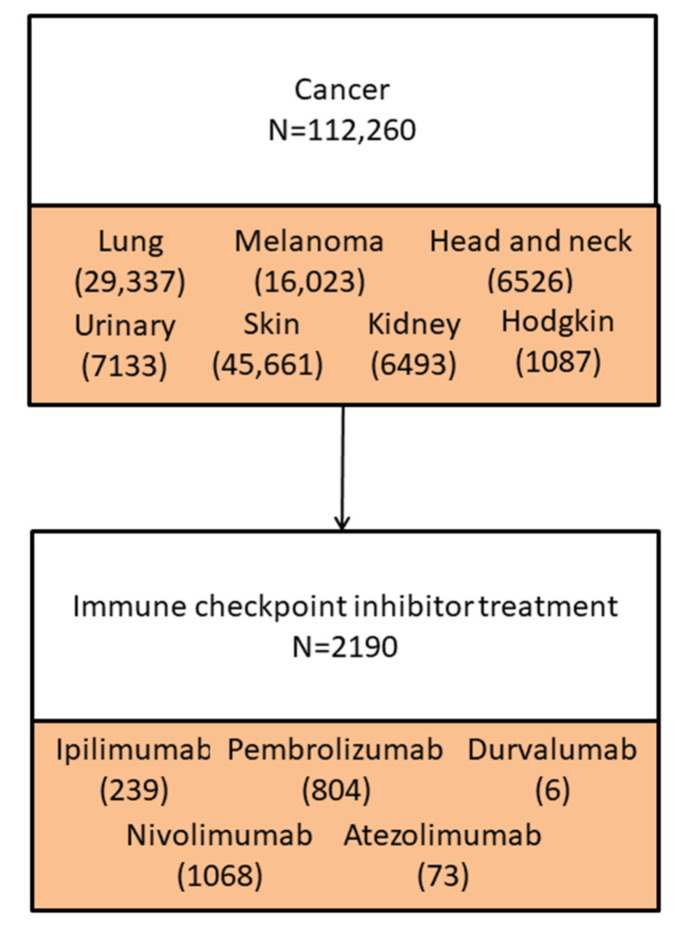
Inclusion of the study population.

**Figure 2 cancers-14-00049-f002:**
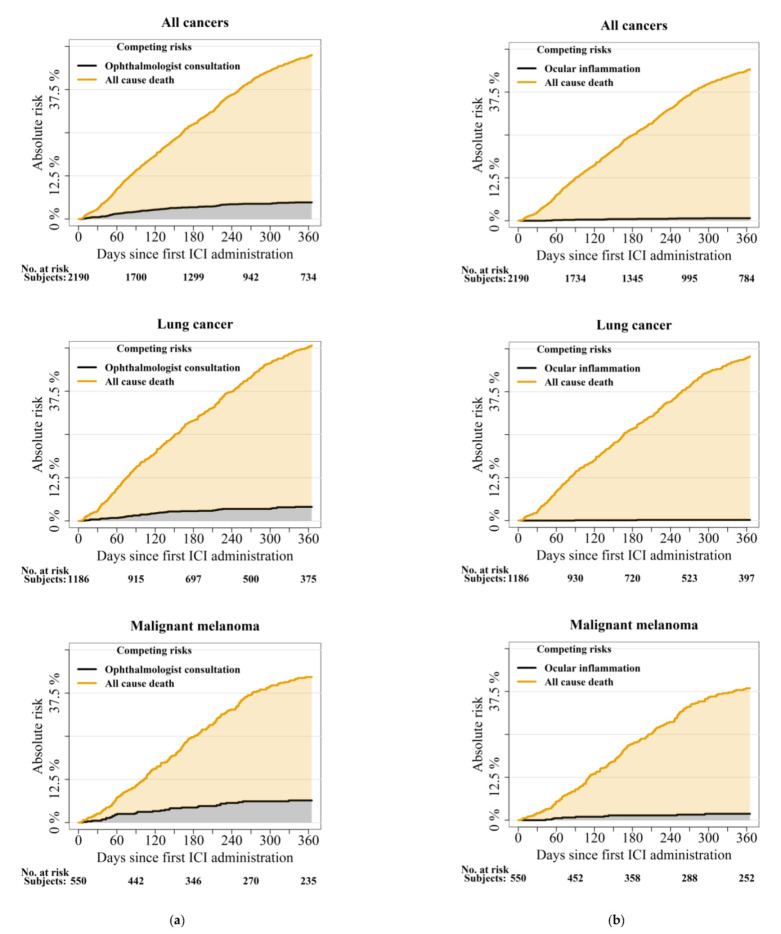
(**a**) Absolute one-year risk of first-time ophthalmologist consultation in patients treated with ICI. (**b)** Absolute one-year risk of first-time ocular inflammation in patients treated with ICI.

**Figure 3 cancers-14-00049-f003:**
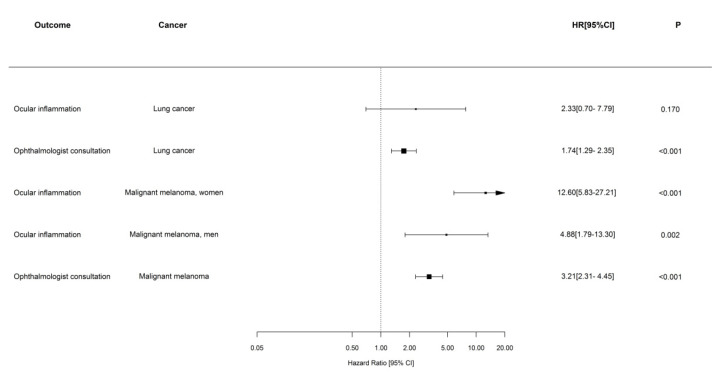
Hazard ratios comparing risk of first-time ophthalmologist consultation and ocular inflammation comparing patients with and without ICI therapy.

**Table 1 cancers-14-00049-t001:** Characteristics of patients with cancer at time of diagnosis.

Patients with Cancer N (%)	112,260 (100)
**Cancer type, N (%)**	
Skin cancer	45,661 (40.7)
Lung cancer	29,337 (26.1)
Malignant cutaneous melanoma	16,023 (14.3)
Head and neck cancer	6526 (5.8)
Urinary tract cancer	7133 (6.4)
Kidney cancer	6493 (5.8)
Hodgkin’s lymphoma	1087 (1.0)
**Age, median [p25–p75]**	**69.4 (59.9–77.5)**
**Male sex, N (%)**	**59,875 (53.3)**
**Medical history, N (%)**	
Diabetes mellitus	10,934 (9.7)
Morbus Bechterew	233 (0.2)
Chronic tissue disease	1647 (1.5)
Inflammatory arthritis	3874 (3.5)
Hypertension	37,055 (33.0)
Chronic kidney disease	4095 (3.6)
Sarcoidosis	250 (0.2)
Syphilis	7 (0.0)
Multiple sclerosis	314 (0.3)
Borrelia infection	95 (0.1)
Human immunodeficiency virus infection	5 (0.0)
Juvenile arthritis	33 (0.0)

**Table 2 cancers-14-00049-t002:** Characteristics for patients treated with immune checkpoint inhibitors at first administration.

Patients Treated with Immune Checkpoint Inhibitor	
**Immune checkpoint inhibitor, N (%)**	
Ipilimumab	239 (10.9)
Pembrolizumab	806 (36.7)
Nivolumab	942 (43.2)
Atezolizumab	73 (3.3)
Durvalumab	6 (0.3)
Ipilimumab + Nivolumab	122 (43.2)
**Cancer type, N (%)**	
Lung cancer	1186 (54.2)
Head and neck cancer	59 (2.7)
Urinary tract cancer	111(5.1)
Malignant cutaneous melanoma	550 (25.1)
Kidney cancer	220 (10.0)
Skin cancer	54 (2.5)
Hodgkin’s lymphoma	10 (0.5)
**Age**. Median [p25–p75]	67 (59–73)
**Male sex, N (%)**	1239 (56.4)
**Medical history, N (%)**	
Diabetes mellitus	224 (10.4)
Morbus Bechterew	NA
Chronic tissue disease	18 (0.8)
Inflammatory arthritis	59 (2.7)
Hypertension	612 (27.9)
Chronic kidney disease	61 (2.8)
Sarcoidosis	11 (0.5)
Syphilis	NA
Multiple sclerosis	5 (0.2)
Borrelia infection	NA
Human immunodeficiency virus infection	NA
Juvenile arthritis	NA

**Table 3 cancers-14-00049-t003:** Absolute risks at 30 days, 6 months, and 1 year after initial ICI-administration.

Outcome	Subgroup (N/Included in Analysis)	N	Absolute Risk (95% CI)	N	Absolute Risk (95% CI)	N	Absolute Risk (95% CI)
		30d	30 Days	182d	6 Months	365d	1 Year
Ocular inflammation	All cancers (2119/2190)	NA	NA	10	0.5 (0.2–0.8)	14	0.8 (0.4–1.2)
Ophthalmologist consultation	All cancers (1648/2190)	12	0.7 (03–1.1)	69	4.5 (3.4–5.5)	92	6.3 (5–7.6)
Uveitis	All cancers (2179/2190)	NA	NA	8	0.4 (0.1–0.7)	9	0.5 (0.2–0.8)
All cause death	All cancers (2190/2190)	74	97.5 (96.8–98.1)	618	75.3 (73.4–77.2)	1023	56.4 (54.0–58.7)
Ocular inflammation	Lung cancer (1144/1186)	NA	NA	NA	NA	NA	NA
Ophthalmologist consultation	Lung cancer (898/1186)	5	0.6 (0.1–1.1)	31	3.7 (2.4–5.0)	42	5.4 (3.8–7.0)
All cause death	Lung cancer (1186/1186)	41	97.4 (96.4–98.3)	369	73.3 (70.6–75.9)	614	52.3 (49.1–55.6)
Ocular inflammation	Malignant melanoma (531/550)	NA	NA	7	1.5 (0.4–2.5)	9	1.9 (0.7–3.2)
Ophthalmologist consultation	Malignant melanoma (428/550)	NA	NA	22	5.6 (3.3–7.9)	31	8.2 (5.4–11.0)
All cause death	Malignant melanoma (550/550)	14	97.9 (96.7–99.1)	124	78.7 (75.0–82.3)	200	62.9 (58.5–67.3)

**Table 4 cancers-14-00049-t004:** Time to event from first ICI administration.

Outcome		N	Median	P25	P75
Ocular inflammation	All cancers	17	140	69	250
Uveitis	All cancers	10	102	54	142
Ophthalmologist consultation	All cancers	108	115	53	223
All cause death	All cancers	1399	212	101	391
Ocular inflammation	Lung cancer	NA	NA	NA	NA
Ophthalmologist consultation	Lung cancer	48	116	69	223
All cause death	Lung cancer	822	206	94	368
Ocular inflammation	Malignant melanoma	11	135	63	272
Ophthalmologist consultation	Malignant melanoma	38	139	54	245
All cause death	Malignant melanoma	294	223	117	265

## Data Availability

The study was based on data from the national administrative databases curated by Statistics Denmark. Access to the data is available after application to Statistics Denmark.

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
