# Peer review of "Immune Checkpoint Inhibitor Treatment and Ophthalmologist Consultations in Patients with Malignant Melanoma or Lung Cancer—A Nationwide Cohort Study"

_cancers, 2021, doi:10.3390/cancers14010049_

Round 1
Reviewer 1 Report
This is a well-written paper by Maria D’Souza and colleagues reporting the effects of immune checkpoint inhibitor (ICI) treatment on frequency of ophthalmologist consultations in patients with malignant melanoma or lung cancer in Danish population.
Authors collected clinical data from 112,260 patients with cancer diagnosed in 2011-2018, 29,337 subjects were with lung cancer, 16,023 154 with malignant cutaneous melanoma, 6,526 with head and neck cancer, 7,133 with urinary 155 tract cancer, 45,661 with skin cancer, 6,493 with kidney cancer and 1,087 with Hodgkin’s 156 lymphoma. Following the cancer diagnosis, 2,190 patients received treatment with ICI. The initial treatment was ipilimumab (10.9%), pembrolizumab (36.7%), nivolumab (43.2%), atezolizumab 167 (3.3%), durvalumab (0.3%) and ipilimumab combined with nivolumab (5.6%). Patients treated with ICI were analyzed for 30 days and 1-year risks for ocular side effects and first ophthalmological contacts. In total, 108 out of 2,190 patients treated with ICI had a first-time ophthalmologist consultation during ICI therapy. The most frequent diagnoses from these consultations were cataract (n=22, 20%), uveal disease (n=10, 9%). The authors concluded that among unselected ICI treated patients, the one year risk of first time ophthalmologist consultations and ocular inflammation reached 6% and 1%. Furthermore, in patients with lung cancer, ICI was associated with higher hazard ratio of first time ophthalmologist consultation (HR=1.74) but was not associated with ocular inflammation. In patients with malignant cutaneous melanoma, ICI was associated with higher hazard ratio of both first time ophthalmologist consultation and ocular inflammation (HR=3.21 and 4.88, respectively).
Data from the Danish population study is presented in four tables and three figures. Tables and figures are very informative and provide great data presentation.
The authors extensively discussed their results in relation to recent scientific literature and clinical trials reports.
Furthermore, the authors report possible limitation of their study that might lead to risk of some biased interpretations (see 4.3. Strengths and limitations).
Paper has 31 references which are relevant to article's subject.
This is an excellent nationwide cohort study, and clinically valuable, especially for those who use ICI in clinical practice or experience ophthalmological issues related to ICI modality. This manuscript provide comprehensive information on this issue.
Taken together, this paper by Maria D’Souza and colleagues represents a worthwhile contribution to the cancer research. I recommend the manuscript for further publication process.
Reviewer 2 Report
This is a well described, written manuscript regarding potential ophthalmic side effects from check point inhibitors. Tables and figures are appropriate.
Regarding ocular inflammation, it correlates with previous publications (see suggested reference below).
The consultation with ophthalmology conclusions can be confusing due to:
- Patients' age: most people will have an ophthalmic consultation at this population age without history of cancer or its treatment.
- Some of comorbidity associations, need regular eye exams regardless the status and treatment of cancer
- If during counseling these patients, they are warned about the potential ocular side effects from ICI, ophthalmic consultation can also increased due to this reason
References: there are several in capital letters; table 1, Syfilis is misspelled, please, correct.
Suggested reference to be added: Ophthalmic Immune-Related Adverse Events of Immunotherapy: A Single-Site Case Series by Jena Kim et al. (Ophthalmology Volume 126, Issue 7, July 2019, Pages 1058-1062)
Reviewer 3 Report
In the present article, the authors attempted to estimate the ocular inflammatory conditions in consecutive patients with malignant melanoma or lung cancer treated with immune checkpoint inhibitors. I have several reservations, my comments are appended as below:
- Reference 6-8- describe the cancer type. I general, while quoting the therapy effectiveness, authors should include the cancer type along with the statistical inference.
- Although immune checkpoints are the central theme, authors should first provide their basis and components in the introduction section. Authors may refer: PMID: 33076303, PMID: 34572799.
- Study population- define clearly the inclusion and exclusion criterion. Were the patients uniformly subjected to treatment?
- Authors should provide the details on patients as sex, median age, etc.
- Authors study comorbidities as diabetes, hypertension. Is it a consequence of CPI treatment or patients were already having these conditions?
- Line 214: although the ocular inflammations are high in both women and men, the HR looks high among women. Do authors have any logical explanation for this?
- Do authors observe any cognitive impairment in patients with ocular inflammations?
- Among the CPI inhibitors used, patients Nivolumab appears to have a high infection rate. Do authors see statistical inference among these treatments (table 2)?
Round 2
Reviewer 3 Report
I congratulate the authors for the modifications. I however suggest taking note of the following points:
- In the first place in the response letter, authors should annotate the line number where details are added, it makes the reviewer's work a bit easy.
- point 1- I could not see the statistical inference as HR, P value-added.
- Line 67-69- instead of just mentioning ICI, authors should mention the type of ICI given.
- Point 2- Unless I am missing, I was unable to track the changes in the main manuscript file. I was expecting a paragraph describing, in brief, the components for the CPI response.
